# Oral Immunization of Larvae and Juvenile of Lumpfish (*Cyclopterus lumpus*) against *Vibrio anguillarum* Does Not Influence Systemic Immunity

**DOI:** 10.3390/vaccines9080819

**Published:** 2021-07-23

**Authors:** My Dang, Trung Cao, Ignacio Vasquez, Ahmed Hossain, Hajarooba Gnanagobal, Surendra Kumar, Jennifer R. Hall, Jennifer Monk, Danny Boyce, Jillian Westcott, Javier Santander

**Affiliations:** 1Marine Microbial Pathogenesis and Vaccinology Laboratory, Department of Ocean Sciences, Memorial University of Newfoundland, St. John’s, NL A1C 5S7, Canada; tmtdang@mun.ca (M.D.); ttcao@mun.ca (T.C.); ivasquezsoli@mun.ca (I.V.); ahossain@mun.ca (A.H.); hgnanagobal@mun.ca (H.G.); 2Department of Ocean Sciences, Ocean Frontier Institute, Memorial University of Newfoundland, St. John’s, NL A1C 5S7, Canada; surendrak@mun.ca; 3Aquatic Research Cluster, CREAIT Network, Department of Ocean Sciences, Memorial University of Newfoundland, St. John’s, NL A1C 5S7, Canada; jrhall@mun.ca; 4Dr. Joe Brown Aquatic Research Building (JBARB), Memorial University of Newfoundland, St. John’s, NL A1C 5S7, Canada; jmonk@mun.ca (J.M.); dboyce@mun.ca (D.B.); 5Fisheries and Marine Institute, Memorial University of Newfoundland, St. John’s, NL A1C 5S7, Canada; jillian.westcott@mi.mun.ca

**Keywords:** lumpfish, cleaner fish, fish larvae, oral vaccine, *Vibrio anguillarum* bacterin, vibriosis, bio-encapsulation, *Artemia salina*

## Abstract

*Vibrio anguillarum*, a marine bacterial pathogen that causes vibriosis, is a recurrent pathogen of lumpfish *(Cyclopterus lumpus).* Lumpfish is utilized as a cleaner fish in the Atlantic salmon (*Salmo salar*) aquaculture in the North Atlantic region because of its ability to visualize and prey on the ectoparasite sea lice (*Lepeophtheirus salmonis*) on the skin of Atlantic salmon, and its performance in cold environments. Lumpfish immunity is critical for optimal performance and sea lice removal. Oral vaccine delivery at a young age is the desired method for fish immunization because is easy to use, reduces fish stress during immunization, and can be applied on a large scale while the fish are at a young age. However, the efficacy of orally delivered inactivated vaccines is controversial. In this study, we evaluated the effectiveness of a *V. anguillarum* bacterin orally delivered to cultured lumpfish and contrasted it to an intraperitoneal (i.p.) boost delivery. We bio-encapsulated *V. anguillarum* bacterin in *Artemia salina* live-feed and orally immunized lumpfish larvae. Vaccine intake and immune response were evaluated by microscopy and quantitative polymerase chain reaction (qPCR) analysis, respectively. qPCR analyses showed that the oral immunization of lumpfish larvae resulted in a subtle stimulation of canonical immune transcripts such as *il8b*, *il10*, *igha*, *ighmc*, *ighb*, *ccl19*, *ccl20*, *cd8a*, *cd74*, *ifng*, and *lgp2*. Nine months after oral immunization, one group was orally boosted, and a second group was both orally and i.p. boosted. Two months after boost immunization, lumpfish were challenged with *V. anguillarum* (7.8 × 10^5^ CFU dose^−1^). Orally boosted fish showed a relative percentage of survival (RPS) of 2%. In contrast, the oral and i.p. boosted group showed a RPS of 75.5% (*p* < 0.0001). *V. anguillar**um* bacterin that had been orally delivered was not effective in lumpfish, which is in contrast to the i.p. delivered bacterin that protected the lumpfish against vibriosis. This suggests that orally administered *V. anguillarum* bacterin did not reach the deep lymphoid tissues, either in the larvae or juvenile fish, therefore oral immunization was not effective. Oral vaccines that are capable of crossing the epithelium and reach deep lymphoid tissues are required to confer an effective protection to lumpfish against *V. anguillarum*

## 1. Introduction

Ectoparasitic sea lice infestations are a serious health issue for wild and cultured Atlantic salmon (*Salmo salar*) [1]. The sea louse (e.g., *Lepeophtheirus salmonis*) feeds on salmonid mucus and tissues leading to physiological and immune failures and may lead to death if left untreated [1,2]. Over the last few decades, the Atlantic salmon industry has used chemical and physical methods to control sea lice infestations [3,4]. The use of chemotherapeutants generates the emergence of resistance in sea lice [3] and enhances negative public opinion on the impacts of aquaculture activity on the environment. Routine mechanical treatments (e.g., abrasion and thermal treatment) used in the salmon industry to physically remove the lice have adverse effects on salmon health and welfare [4]. Currently, multiple non-medicinal methods are used in different combinations (e.g., curtains and skirts, anti-lice diets, laser, water jets, and ultrasound) as part of an integrated pest management (IPM) strategy [5]. Another current strategy to control sea lice in the North Atlantic farms is the use of cleaner fish species as a biological control [4,6]. The use of cleaner fish is considered an eco-friendly alternative, as it can reduce the use of chemotherapeutants, and it is less stressful to farmed fish [7]. In Atlantic Canada, both lumpfish (*Cyclopterus lumpus*) and cunner (*Tautogolabrus adspersus*) are used as cleaner fish [8,9]. However, lumpfish is the preferred species for the biocontrol of sea lice due to its performance in cold environments and biomass availability [10,11]. In contrast to other cleaner fish species, lumpfish actively remove sea lice from farmed salmon in cold environments, and they have been industrialized in the North Atlantic region [12,13,14,15]. Use and demand for lumpfish in salmon farms in Ireland, the United Kingdom (UK), Norway, the Faroe Islands, Iceland, and Canada have increased in recent years [13,16]. For example, 11.8 and 0.8 million juveniles were used in Norway and in the UK in 2015, respectively [17]. In 2018, approximately 30 million juvenile lumpfish were used in Norway [18]; in the UK, approximately 6 million lumpfish were used; in Iceland, approximately 3.5 million were used [19]; and approximately 300 thousand lumpfish were used in Ireland [20]. In Canada, the use of lumpfish is a very recent practice, and approximately 1 million lumpfish were used in Atlantic Canada in 2019 [19].

Similar to other finfish species, lumpfish is susceptible to different types of bacterial pathogens such as *Aeromonas salmonicida*, *Pasteurella* spp., *Tenacibaculum* spp., *Pseudomonas anguilliseptica*, and *Moritella viscosa* [21,22]. Vibriosis, typically caused by *Vibrio anguillarum*, is a prevalent pathogen in lumpfish in Newfoundland [23], but others such as *V. ordalii*, *V. salmonicida*, and *V. splendidus* also infect lumpfish elsewhere [12,24]. *V.*
*anguillarum* is a Gram-negative, motile, rod-shaped bacterium and a ubiquitous marine pathogen of invertebrates and vertebrates [25,26]. Formalin-killed *V. anguillarum* (bacterins) serotypes O1 and O2 mixed with adjuvants are often used to formulate vaccines for different fish species, including Atlantic salmon, Atlantic cod (*Gadus morhua*) [27], gilthead seabream (*Sparus aurata*), and lumpfish [28]. Intraperitoneal (i.p.) injection is the most common method of vaccine delivery [29], but it cannot be applied to small fish. An oral-immersion vaccine is ideal for fish immunization, as it is needle-free and causes minimum stress during application [30]. However, oral vaccine delivery to small fish or larvae has been evaluated with controversial results [31]. Vaccine bio-encapsulation strategies, such as the used of *Artemia*
*salina* fed with bacterin, can increase the delivery of the vaccine to small lumpfish larvae by increasing intake and can directly deliver the antigen to the fish gut [32,33,34]. 

Here, we evaluated the efficacy of a *V. anguillarum* bacterin bio-encapsulated in *A. salina* delivered as live-feed to lumpfish larvae. We determined that the oral *V. anguillarum* vaccine delivered in *A. salina* live-feed reached the lumpfish gut. Real-time quantitative polymerase chain reaction (qPCR) analyses of immune-relevant genes revealed that oral immunization modestly immune stimulated the lumpfish larvae. Nine months later, a lumpfish group was orally boosted, and an independent second group was orally and i.p. boosted. Two months later, the immunized fish groups were i.p. challenged with ten times the lethal dose 50 (LD_50_) of *V. anguillarum* (7.8 × 10^5^ colony forming units (CFU) dose^−1^). Oral immunization of lumpfish delayed mortality but did not confer protective immunity against the *V. a**nguillarum* challenge, which contrasts with the i.p. vaccination, which protected the fish against the lethal infection.

## 2. Materials and Methods

### 2.1. Vibrio anguillarum J360 Culture Conditions

*V. anguillarum* J360 serotype O2 (NCBI IDs: Chromosome 1 CP034672; Chromosome 2 CP034673; and plasmid CP034674), a local lumpfish isolate, was used in this study [23]. *V. anguillarum* J360 was grown in 3 mL of tryptic soy broth (TSB, Difco) for 24 h at 28 °C with aeration (180 rpm, in an orbital shaker). Bacterial growth was monitored using spectrophotometry (Genesys 10 UV spectrophotometer, Thermo Fisher Scientific Inc., Waltham, MA, USA) and by plating to determine the CFU mL^−1^. When the optical density (OD600 nm) reached ~0.7 (1 × 10^8^ CFU mL^−1^), the cells were harvested by centrifugation at 4200× *g* for 10 min at room temperature. The cell suspension was washed twice with phosphate-buffered saline (PBS; 136 mM NaCl, 2.7 mM KCl, 10.1 mM Na_2_HPO_4_, 1.5 mM KH_2_PO_4_; pH 7.2) [35] at 4200× *g* and resuspended in 300 μL of PBS 1X (~1 × 10^10^ CFU mL^−1^). The final inoculum was serially diluted (1:10), and the concentration was determined by plate counting [36] in TSB supplemented with 1.5% bacto agar (TSA).

### 2.2. Bacterin Preparation

The bacterin preparation was conducted according to established protocols and quantified using flow cytometry enumeration [37] with modifications. First, *V. anguillarum* J360 was grown in 500 mL of TSB supplemented with a final concentration of 150 μM 2,2-dipyridyl (Sigma-Aldrich, St. Louis, MO, USA) at 28 °C with aeration (180 rpm, in an orbital shaker) to induce synthesis of the iron-regulated outer membrane proteins (IROMPs) [38]. Bacterial growth was monitored spectrophotometrically until it reached ~1 × 10^8^ CFU^−1^. *V. anguillarum* cells were harvested using centrifugation (4200× *g* at 4 °C for 10 min) and washed twice with PBS. *V. anguillarum* cells were fixed with buffered formalin 6% (Sigma) at room temperature for 3 d with gentle agitation in a rocker shaker. Cell viability was determined each day by plating onto TSA. Formalin was removed using centrifugation, and cells were then dialyzed (Spectrum™ Spectra/Por™ dialysis membrane 12–14,000 Dalton molecular weight cut-off, Thermo) in 2 L of PBS at 6 °C for 3 d with agitation. *V. anguillarum* bacterins were quantified using the BD FACS Aria II flow cytometer (BD Biosciences, San Jose, CA, USA) and BD FACS Diva v7.0 software as described previously [39]. Bacterin cells (1 × 10^10^ CFU mL^−1^) were stored at 4 °C until use.

### 2.3. V. anguillarum Bacterin Fluorescent Labeling

*V. anguillarum* bacterin was labeled with 5-([4,6-dichlorotriazinyl] amino) fluorescein hydrochloride (DTAF; Sigma) according to previously described protocols [40] with minor modifications. First, *V. anguillarum* bacterin (~1 × 10^10^ CFU mL^−1^) was centrifuged at 4200× *g* for 10 min and then resuspended in 950 µL of bicarbonate buffer (0.1 M, pH 9). Following that, the cells were mixed with 50 µL of DTAF solution (100 µg in dimethyl sulfoxide (DMSO); Sigma) and incubated overnight at 4 °C in dark conditions. After incubation, the bacterin cells were centrifuged (4200× *g* for 10 min) and washed three times with bicarbonate buffer and were finally resuspended in PBS and kept at 4 °C until use. 

Next, we describe the optimization of *V.*
*anguillarum* bacterin bio-encapsulation in *A. salina*. To optimize bacterin bio-encapsulation in *A. salina*, we used the method described by Campbell et al. [32] with modifications. Additionally, we developed a semi-quantitative method to estimate the levels of bacterin bio-encapsulation in *A. salina*. First, *A. salina* nauplii were hatched from cysts according to the supplier’s instructions (INVE, Salt Lake City, UT, USA) (Appendix A). After the *A. salina* nauplii hatched (~20 h at 20 °C), nauplii were washed with seawater for 30 min (Appendix A). *A. salina* cultures were nutritionally supplemented with Ori-One (Skretting, Fontaine les Vervins, France) and Ori-Green (Skretting, Fontaine les Vervins, France) commercial dry microalgae extract at a ratio of 1:1 (Ori-One:Ori-green) for 3 h at 20 °C. To determine the optimal time for bacterin bio-encapsulation in A. salnina, supplemented *A. salina* nauplii were inoculated into 6 well plates with a 3 mL total volume per well at a density of 1000 nauplii mL^−1^. Additionally, a separate plate was inoculated with non-supplemented *A. salina* nauplii to determine the possible effect of nutrient supplementation on bio-encapsulation. Both groups were inoculated with DTAF-labeled *V. anguillarum* bacterin (5 × 10^7^ cells mL^−1^). The control group was mock inoculated with seawater and used to evaluate autofluorescence. The nauplii were incubated at 20 °C for 48 h to determine the optimal time for bacterin bio-encapsulation (Appendix A). *A. salina* samples (1 mL) were collected at 1, 3, 5, 24, 36, and 48 h, and fixed in 10% buffered formalin. The presence of *V. anguillarum* inactivated bacterin was examined and counted using confocal microscopy (Nikon Eclipse Ti, Melville, NY, USA) to determine the number of *A. salina* containing 0%, 25%, 50%, 75%, 100% bacterin (Appendix A). After determining the optimal time for bacterin bio-encapsulation in *A. salina* nauplii, we evaluated the bio-encapsulation stability at 6 °C for 6 d (Appendix A). A. salina nauplii were supplemented with Ori-One, Ori-Green, and DTAF-labeled *V. anguillarum* bacterin at 20 °C for 3 h and then placed at 6 °C for 6 d. The *A. salina* control group was mock inoculated with seawater. *A. salina* samples were collected each day and fixed with buffered 10% formalin. The levels of bio-encapsulation in *A. salina* were determined using confocal microscopy (Appendix A). DTAF-labeled *V. anguillarum* inactivated bacterin bio-encapsulated in *A. salina* nauplii were used to feed the lumpfish larvae (Appendix A) and to determine the presence of *V. anguillarum* bacterin in the larvae gut compared to the non-orally immunized fish (Appendix A). Fifty lumpfish larvae were orally immunized and maintained at 6 °C for 24 h. The larvae gut was observed at 0, 0.5, 1, 2, 4, 6, and 24 h post-oral immunization using epi-fluorescence microscopy (Optika, Italy) (Appendix A).

### 2.4. V. anguillarum Bacterin Bio-Encapsulation in A. salina

Based on the vaccine bio-encapsulation optimization results, larger volumes of *A. salina* were prepared for lumpfish larvae immunization. *A**. salina* nauplii were hatched and washed with seawater for 30 min and placed in 20 L buckets containing 15 L of seawater. *A**. salina* nauplii were maintained at a density of ~2.5 million *A**. salina* per liter. *A**. salina* cultures were enriched with nutritional supplements derived from microalgae, Ori-One (Skretting, France) and Ori-Green (Skretting, France) according to the manufacturer’s instructions. *A**. salina* cultures were inoculated with *V. anguillarum* bacterin (10^7^ cells mL^−1^) and incubated at 20 °C for 3 h with aeration. Controls were mock inoculated with PBS. After 3 h of enrichment and bacterin bio-encapsulation, the *A**. salina* cultures were maintained at 6 °C for 6 d under constant light. These cultures were used for the daily oral immunization of the lumpfish larvae (Appendix A). 

### 2.5. Aquafeed Coating with V. anguillarum Bacterin

Commercial dry feed was coated with dry *V. anguillarum* bacterin to orally boost immunized fish. Ficoll, a non-toxic polymer, was used as a cryoprotectant for bacterial lyophilization [41]. Ficoll also serves as an antigen and adjuvant carrier [41,42,43,44,45,46]. To freeze-dry the bacterin, a formalin-killed *V. anguillarum* (2 × 10^9^ CFU mL^−1^) suspension was mixed with Ficoll solution (20% Ficoll400 (GE Healthcare, Sweden), 300 mM NaCl) at a 1:1 ratio to prevent cell lysis during lyophilization. The cells were lyophilized (Edwards super module E2-M5, Edwards, UK) for 3 days. The bacterin powder was mixed with 3–4 mm commercial dry pellet (Skretting-Europa 15: crude protein (55%), crude fat (15%), crude fiber (1.5%), calcium (3%), phosphorus (2%), sodium (1%), vitamin A (5000 IU kg^−1^), vitamin D (3000 IU kg^−1^), and vitamin E (200 IU kg^−1^)) at the ratio of 0.9 g bacterin per 100 g aquafeed. After mixing the feed with the bacterin powder, a layer of cod liver oil was added (3 mL 100 g feed^−1^), and the feed was then dried at room temperature to complete the coating process. The coated feed was stored at 4 °C until used.

### 2.6. Fish Culture Conditions

All animal protocols required for this research were approved by the Institutional Animal Care Committee and the Biosafety Committee at Memorial University of Newfoundland (MUN). Experiments were conducted under protocols #18-01-JS, #18-03-JS, and biohazard license L-01. Lumpfish egg masses were in vitro fertilized and maintained in 5 L buckets containing UV treated (300 mW cm^−^^2^), filtered, flow-through seawater (33 ppt) at 10–10.5 °C, with 95–110% air saturation, and held under ambient photoperiod (spring–summer) at the Joe Brown Aquatic Research Building (JBARB), Department of Ocean Sciences, Memorial University of Newfoundland, Canada. After embryo development was complete, the larvae were hatched and maintained in the same seawater conditions and air saturation at 10 °C until full egg yolk sac absorption and the establishment of independent feeding was achieved (Appendix A). 

### 2.7. Lumpfish Immunization Assays

Lumpfish larvae were stocked in 4 tanks at a density of 2000 larvae per tank (20 L) with UV treated (300 mW cm^−^^2^), filtered, flow-through seawater at 8–10 °C, and with 95–110% air saturation (Figure 1). Two tanks containing lumpfish larvae (1-week post-hatch (wph)) were orally immunized with the bio-encapsulated *V. anguillarum* vaccine daily for 4 weeks. Two tanks fed containing only *A. salina* served as controls (Figure 1). Lumpfish larvae were fed with *A. salina* nauplii or bio-encapsulated vaccine 3 times per day (350 mL of *A. salina* culture per 2000 larvae) (Appendix A). Thereafter, the fish were fed *A**. salina* for additional 10 d and then fed with dry pelleted diet daily (0.75–2% body weight). Whole larvae pool samples were collected at 0, 2, and 4 weeks post-immunization (wpi) (Figure 1). At each time point, triplicate pools of 5–10 larvae were sampled from each tank and placed in a 1.5 mL RNase-free tube containing 300 μL of TRIzol Reagent (Invitrogen, Waltham, MA, USA), flash-frozen in liquid nitrogen, and stored at −80 °C until processing. After 8 weeks, the juvenile lumpfish were transferred into eight 500 L tanks (Figure 1). The lumpfish were fed an assorted size pelleted diet for the remainder of the experiment (15 months). Nine months post-hatch, two tanks with 100 fish each (~72.1 ± 30 g) were orally boost immunized using commercial pellets coated with the *V. anguillarum* bacterin. Lumpfish were starved for 24 h pre-oral vaccination. Lumpfish were fed with *V. anguillarum* bacterin coated dry pellets three times (every 2 weeks for 3 days) at 0.75% body weight (Figure 1). Two control tanks were mock-orally boosted with ficoll (vaccine vehicle) coated pellets (Figure 1). Two independent groups orally boosted were additionally i.p. boosted at 40 wph (~145 ± 31.8 g) with *V. anguillarum* bacterin (6.3 × 10^8^ cells dose^−1^). Control groups were mock-orally and i.p. boosted with the respective vaccine vehicle (Figure 1). Two months later, the animals (~132–244 g) were transferred to the AQ3 aquatic biocontainment facility at the Cold-Ocean Deep-Sea Research Facility (CDRF) for challenge assays.

### 2.8. V. anguillarum J360 Challenge Assays in Lumpfish

The challenge assays were conducted at the CDRF AQ3 biocontainment facility under the Institutional Animal Care Committee approved protocol (18-02-JS) and established protocols [28]. First, after transfer to the AQ3 biocontainment facility, lumpfish were acclimated for 1 week under optimal conditions prior to the commencement of the challenge. Lumpfish were challenged by an i.p. injection with 7 times the lethal dose 50 (7.8 × 10^5^ CFU dose^−1^) of *V. anguillarum* J360. Fish survival was monitored for 30 days post-challenge. The relative percent of survival (RPS) of the control and vaccinated fish was calculated using the formula: RPS = [1 − (% mortality in vaccinated fish/% mortality in control fish)] × 100 [47].

### 2.9. Total RNA Extraction 

RNA was extracted from the larvae pools pre-immunized (*n* = 3 individual pools of 10 larvae each), 2 wpi (*n* = 3 pools of 10 larvae each), and 4 wpi (*n* = 3 pools of 5 larvae each). RNA was also extracted from the larvae post mock immunization(control) at 2 and 4 wpi. Lumpfish larvae pools, previously flash frozen in a 1.5 mL RNase-free tube containing 300 μL of TRIzol reagent (Invitrogen), were homogenized using a micro-tube homogenizer (ThermoFisher Scientific, Waltham, MA, USA). An additional 700 μL of TRIzol was added to the tube, and the extractions were then completed following the manufacturer’s instructions. The TRIzol extracted samples were then purified using the RNeasy^®^ Mini Kit (QIAGEN, Mississauga, ON, Canada) following the manufacturer’s instructions. RNA samples were treated with 2 U mL^−1^ of TURBO DNase (TURBO DNA-free™ Kit, Invitrogen) following the manufacturer’s instructions for the complete digestion of genomic DNA and the removal of the remaining DNase and divalent cations, such as magnesium and calcium. Purified RNA samples were quantified and evaluated for purity using a Genova Nano spectrophotometer (Jenway, Staffordshire, UK) and evaluated for integrity using 1% agarose gel electrophoresis [35]. A PCR test was conducted using the *60S ribosomal protein L32* (*rpl32*) reference gene primers and the RNA as a template to rule out the presence of DNA. 

### 2.10. cDNA Synthesis and qPCR Parameters

cDNA was synthesized in 20 μL reactions from 1 μg of RNA using SuperScript IV VILO Master Mix (Invitrogen) following the manufacturer’s instructions. PCR amplifications were performed in 13 μL reactions using 1X Power SYBR Green PCR Master Mix (Applied Biosystems, Waltham, MA, USA), 50 nM of both the forward and reverse primers, and the indicated cDNA quantity. Amplifications were performed using the QuantStudio 6 Flex Real-Time PCR system (384-well format) (Applied Biosystems, Waltham, MA, USA). The real-time analysis program consisted of 1 cycle of 50 °C for 2 min, 1 cycle of 95 °C for 10 min, 40 cycles of 95 °C for 15 s, and 60 °C for 1 min, with fluorescence detection at the end of each 60 °C step and followed by dissociation curve analysis.

### 2.11. qPCR Primer Quality Assurance Testing

All primer pairs for the transcripts of interest (TOIs) and the endogenous control transcripts were designed and quality control (QC) tested using the larvae RNA samples generated herein. cDNAs were synthesized from the individual pooled larvae RNA samples, including pre-immunized control, mock-immunized control (2 and 4 wpi), and immunized larvae (2 and 4 wpi). The control and immunized cDNA samples were independently pooled and used for primer quality evaluation. To calculate amplification efficiencies for each primer pair [48], standard curves were generated for both cDNA pools (control and immunized) using a 5-point 1:3 dilution series starting with cDNA representing 10 ng of input total RNA. The reported efficiencies represent an average of the two values (Appendix A). Each primer pair was also tested to ensure that a single product was amplified and that there was no primer dimer present in the no-template control. Finally, amplicons were electrophoretically separated on 2% agarose gels and compared using a 1 kb plus ladder (Invitrogen) to verify that the correct size fragment was amplified.

Eighteen TOIs were well expressed in larvae, and as such, amplification efficiencies could be calculated using larvae cDNA template (see fluorescence threshold cycle (C_T_) values for studies 1 to 3; Appendix A). However, fourteen of these transcripts were expressed at low levels in larvae (see C_T_ values for studies 4 to 6; Appendix A). For the latter TOIs, technical replicates and spacing were acceptable over the first three points of the cDNA dilution series. As the experimental input cDNA amount (8 ng) for these TOIs lies within the first 2 dilutions, these assays were deemed acceptable for analysis in larvae. However, for these fourteen transcripts, the amplification efficiencies reported in Appendix A and inputted into the QuantStudio Real-Time PCR Software (version 1.3) (Applied Biosystems) were those that had been previously generated for head kidney samples. 

### 2.12. Endogenous Control (Normalizer) Selection

Expression levels of the TOIs were normalized to the transcript levels of the two endogenous controls. To select these endogenous controls, 5 transcripts (*rpl32*, *elongation factor 1-alpha* (*ef1a*), *eukaryotic translation initiation factor 3 subunit D* (*etif3d*), *polyadenylate-binding protein 1a* (*pabpc1a*), and *polyadenylate-binding protein 1b* (*pabpc1b*)) were analyzed. First, the C_T_ values of all 27 samples in the study were measured (in duplicate) for each of these transcripts using cDNA representing 3.25 ng of input total RNA and then analyzed using geNorm [49]. Based on this analysis, *rpl32* (geNorm M = 0.169) and *etif3d* (geNorm M = 0.177) were selected as the two endogenous controls.

### 2.13. Experimental qPCR Analyses

To study the effects of oral immunization with the *V. anguillarum* bacterin bio-encapsulated in *A. salina* on the immunome of lumpfish larvae, expression levels of 32 TOIs with immune-relevant functional annotations were assessed (Appendix A). Individual larvae pools (*n* = 27 pools) were subjected to qPCR analyses (Figure 1). In the qPCR analyses, cDNA representing 3.25 ng (study 1 to 3, see Appendix A) and 8 ng (study 3 to 6, see Appendix A) of input RNA was used as a template in the PCR reactions. The input RNA concentration was increased to 8 ng due to lower expression transcript levels in the larvae. In each qPCR study, expression levels of a given transcript were measured across two plates. On each plate, the TOIs and endogenous controls were tested in triplicate, and a no-template control was included for every sample. A plate linker sample was also included to ensure that there was no plate-to-plate variability. The relative quantity (RQ) of each transcript was determined using the QuantStudio Real-Time PCR Software (version 1.3), with normalization to both the *rpl32* and *etif3d* transcript levels and with the amplification and efficiencies incorporated. For each TOIs, the sample with the lowest normalized expression (mRNA) level was set as the calibrator sample (i.e., assigned an RQ value = 1) (Appendix A). Additionally, the transcript expression levels were determined using the comparative 2^−ΔΔCt^ method [50] with two reference genes [51] (Appendix A). 

### 2.14. Statistical Analysis

All data are expressed as the mean ± standard error (SE). Assumptions of normality and homogeneity were tested for variances. A one-way ANOVA followed by Tukey’s multiple comparison post hoc test was used to determine significant differences between the survival of the control and infected groups. The Kaplan–Meier estimator was applied for the estimation of the survival fractions after the *V. anguillarum* challenges, and the log-rank test was used to identify differences between treatment groups (*p* < 0.0001). A two-way ANOVA was used to analyze the gene expression data followed by Sidak’s multiple comparisons post hoc test to identify significant differences between each treatment in the control and immunized groups at each time point (2 weeks and 4 weeks). All statistical tests were performed using Graphpad Prism version 8.0 (Graphpad Software, USA, www.GraphPad.com (accessed on 15 June 2021)), and *p*-values < 0.05 were considered statistically significant. 

## 3. Results

### 3.1. Bio-Encapsulation of V. anguillarum Bacterin in A. salina Nauplii

We established a semi-quantitative method to estimate the levels of *V. anguillarum* bacterin bio-encapsulation in *A. salina* (Figure 2A). Autofluorescence was ruled out using *A. salina* nauplii inoculated with nutritional supplements (Ori-One and Ori-Green) (Figure 2B). Using this method, we determined that approximately 100% of the *A. salina* nauplii reached the maximum capacity for *V. anguillarum* bacterin bio-encapsulation 3 h post-inoculation at 20 °C in both the absence (Figure 2C) or presence (Figure 2D) of nutritional supplements. In the absence of supplements, a significant decrease in the percentage of the *A. salina* nauplii with 100% bio-encapsulation levels was observed 24 h post-inoculation, which gradually declined thereafter (Figure 2C). In the presence of nutritional supplements, a significant decrease occurred 36 h post-inoculation (Figure 2D). We determined that the optimal *V. anguillarum* bacterin bio-encapsulation in *A. salina* method is the presence of nutritional supplementation with a bio-encapsulation time of 3 h at 20 °C. As we wanted to produce a bio-encapsulated vaccine batch that could be used for several days, we evaluated the stability of the *V. anguillarum* bacterin in the intestine of *A. salina* nauplii post bio-encapsulation at 6 °C (Appendix A). This temperature was chosen as it is similar to the water temperature at which lumpfish are cultured. We determined that the bacterin concentration in *A. salina* nauplii remained stable for up to 6 d post-inoculation (Figure 2E). DTAF-labeled *V. anguillarum* bacterin bio-encapsulated in *A. salina* nauplii were then fed to lumpfish larvae and fluorescence microscopy was used to determine if the *V. anguillarum* bacterin reached the larvae gut (Appendix A). Fluorescence microscopy demonstrated the presence of the *V.*
*anguillarum* bacterin in the gut of the lumpfish larvae after 6 and 24 h, whereas fluorescence was not detected in the gut of lumpfish larvae who had been fed *A. salina* nauplii only (Figure 3). 

### 3.2. Transcript Expression Profile of the Immunome of Orally Immunized Lumpfish Larvae

Expression levels of the transcripts related to the innate and adaptive immune response were measured in pre-immunized larvae and at 2 and 4 wpi with the *V. anguillarum* bacterin bio-encapsulated in *A. salina* nauplii. The transcript expression levels were analyzed using both the 2^−ΔΔCt^ (Figure 4 and Figure 5) and the RQ methods (Appendix A). Both methods showed similar results, with the exception of the statistical significance demonstrated for *toll-like receptor 7* (*tlr7*) and *immunoglobulin heavy chain b* (*ighb*) (Figure 4, Figure 5 and Figure 6; Appendix A). Transcript expression levels were compared statistically in orally immunized compared to control larvae at 2 and 4 wpi only. Comparisons over time could not be assessed due to developmental and considerable size differences of the larvae over time. Significant up-regulation of *interleukin 8b* (*il8b*) (Figure 4A and Appendix A), *immunoglobulin heavy chain a* (*igha*), *ighb* and *immunoglobulin mu heavy chain c* (*ighmc*) (Figure 5A and Appendix A), *chemokines* (*ccl19* and *ccl20*) (Figure 5B and Appendix A), *cluster of differentiation 8 alpha* (*cd8a*) and *HLA class II histocompatibility antigen gamma chain* (*cd74*) (Figure 5C and Appendix A), *interferon-gamma* (*ifng*) (Figure 6 and Appendix A), and *ATP-dependent RNA helicase lgp2* (*lgp2*) (Figure 6 and Appendix A) occurred at 2 wpi. Significant up-regulation of *interleukin 10* (*il10*) occurred at 4 wpi (Figure 4A and Appendix A). Significant downregulation of *lymphocyte antigen 6 family member G6F* (*ly6g6f*) (Figure 6 and Appendix A) and *ccl20* (Figure 6B and Appendix A) occurred at 2 and 4 wpi, respectively. There were no significant differences in the expression levels of the remaining transcripts at either time point.

### 3.3. Vaccine Challenge

Immunized lumpfish were challenged at 45 wpi with 7 times the LD_50_ dose for *V. anguillarum* J360 (7.8 × 10^5^ CFU dose^−1^) (Figure 1). Mortality in lumpfish who had been orally mock immunized as larvae and then mock-orally boosted as juveniles started at 3 days post-challenge, with 100% mortality by day 10 post-challenge. Mortality in lumpfish that had been orally immunized as larvae and then orally boosted as juveniles was delayed and started at 3 days post-challenge, with a final RPS of 2% (Figure 7A). Mortality in lumpfish that had been mock orally immunized as larvae and then both mock orally and i.p. boosted as juveniles started at 7 days post-challenge, with 100% mortality by 20 days post-challenge (Figure 7B). Lumpfish that had been orally immunized as larvae and then both orally and i.p. boosted as juveniles survived the i.p. challenge with *V. anguillarum*, with a RPS of 76.5% (*p* < 0.0001) (Figure 7B).

## 4. Discussion

Vibriosis is one of the most common bacterial diseases affecting lumpfish aquaculture [39]. As mentioned previously, immunization of fish at an early age with a needle-free vaccine and with minimal stress during immunization is the ideal vaccine delivery method for finfish aquaculture [30]. However, the effectiveness of bath and oral inactivate vaccine delivery to small fish or larvae has been evaluated with varied results [52]. Commercial bath vaccines have been used in lumpfish with low effectiveness against local *V. anguillarum* isolates [28]. However, the efficacy of an orally administered vaccine in lumpfish during the early-life stages remains unknown. Here, we evaluated the efficacy of an orally delivery *V. anguillarum* bacterin bio-encapsulated in *A. salina* nauplii in lumpfish larvae. The molecular immune response to oral immunization was evaluated in larvae by qPCR. Approximately 9 months after the initial oral immunization, lumpfish were then either orally or both orally and i.p. boosted, and the effectiveness of the vaccines was evaluated by assessing the RPS after a lethal i.p. *V. anguillarum* challenge.

First, we evaluated the *V. anguillarum* bacterin uptake in *A. salina* nauplii and, thereafter this bio-encapsulated bacterin, in the gut of lumpfish larvae. Our observations indicated that the *V. anguillarum* bacterin was fully bio-encapsulated by the *A. salina* nauplii after 3 h and was maintained for at least 6 d at 6 °C (Figure 2). Similar results were observed by Campbell et al. [32], where the *V. anguillarum* bacterin showed maximum bio-encapsulation after 1 h or 2 h using 1.5 × 10^7^ CFU mL^−1^ or 1.5 × 10^6^ cells mL^−1^, respectively. Vaccine bio-encapsulation in *A. salina* nauplii protects the antigens from the intestinal tract of the fish and facilitates the recognition of the antigens using the macrophages in the mucosal layer of the hindgut [53]. The effectiveness of protecting the antigen from gastrointestinal digestion and its delivery to the hindgut of fish larvae has been demonstrated in previous studies [33,53]. Here, we confirmed the presence of the *V. anguillarum* bacterin in the *A. salina* nauplii and in the gut of fish larvae 6 h post-oral immunization (Figure 3). These results validated the internalization of the *V. anguillarum* bacterin in the lumpfish gut.

The expression profiles of 32 TOIs related to innate and adaptive immunity were evaluated in larvae pre-immunization and at 2 and 4 wpi. In pre-immunized larvae, we did not see any expression of the TOIs, which was expected. When considering orally immunized compared to larvae who had been orally mock immunized at 2 and 4 wpi, there were no significant differences in the levels of pro-inflammatory cytokines (*tnfa*, *il1b*, *il8a*; Figure 4A and Appendix A), toll-like receptors (*tlr3*, *tlr5a*, *tlr5b*; Figure 4B and Appendix A), immunoglobulin heavy chain transcripts (*ighma*, *ighd*; Figure 5A and Appendix A), interferon-induced effectors (*mxa*, *mxb*, *mxc*; Figure 5B and Figure 7B), cluster of differentiation transcripts (*cd4a*, *cd4b*; Figure 5C and Appendix A), or other immune-related transcripts (*cox2*, *irf7*, *lgp2*, *stat1*, *rsad2*, *hamp*, *saa5*; Figure 6 and Appendix A). In contrast, levels of *il**8b*, *igha*, *ighmc*, *ighb*, *ccl19*, *ccl20*, *cd8a*, *cd74*, *if**n**g*, and *lgp2* were significantly upregulated, and levels of *ly6g6f* and *tlr7* were significantly downregulated at 2 wpi (Figure 4A, Figure 5 and Figure 6, Appendix A). Levels of *il10* were significantly upregulated, and levels of *ccl20* were significantly downregulated at 4 wpi (Figure 4A and Figure 5B, Appendix A). These results indicate that 35 d old lumpfish larvae are not highly immune stimulated by oral immunization, suggesting that the interaction between the lymphoid tissues and the vaccine was not enough to trigger adaptive immune protection. 

*il8* and *ifng* play important roles in the recruitment of monocytes and neutrophils to sites of inflammation. Whereas *il**10* acts as anti-inflammatory cytokine and as such, plays a crucial role in the regulation of the inflammatory response [54]. Although there are studies on the expression of *il**8* and *il**10* in fish, the role of these interleukins in the early developmental stages of lumpfish is still unknown. In the current study, in orally immunized larvae compared to larvae who had been orally mock immunized, *il8* and *ifng* were significantly upregulated at 2 wpi, *il10* significantly upregulated at 4 wpi (Figure 4A, Figure 5 and Figure 6; Appendix A). Similar expression profiles for *ifng* and *il10* have been observed in Atlantic salmon that had been infected with the salmonid alphavirus subtype-3 (SAV-3) [55] or immunized with the *A. salmonicida* vaccine [56]. *lgp2* is a member of the RLR family that participates in the recognition of viral RNA pathogen-associated molecular patterns (PAMPs) in the cytoplasm and induces the synthesis of *if**n**g* [57,58,59,60]. Similarly, our results showed that *lgp2* and *if**n**g* were significantly upregulated at 2 wpi (Figure 6 and Appendix A). The results suggest that the oral immunization of lumpfish larvae triggers an innate immune response that is later regulated via the canonical anti-inflammatory cytokine *il10*.

TLRs play an important role in early innate and adaptive immunity by detecting PAMPs in bacteria and viruses [61,62,63]. TLRs activate the transcription factor NF-κB, resulting in the production of several pro-inflammatory cytokines such as *il**1b*, *tnfa*, *il8*, *il10*, *il6*, *il12*, *il17*, *if**n**g*, and the *tumour necrosis factor* (*tnf*) [64,65]. Expression levels of *tlr3*, *tlr5,* and *tlr7* were not significantly different in orally immunized compared to larvae who had been orally mock immunized at either time point (Figure 4B), however *tlr7* was significantly downregulated at 2 wpi in the RQ statistical analysis (Appendix A). These results suggest that the vaccine does not induce a full immune response in larvae. 

*ly6g6f* is a member of the superfamily lymphocyte antigen-6 (Ly6)/urokinase-type plasminogen activator receptor (uPAR) [66]. Ly6/uPAR proteins have functions in cell proliferation, migration, cell–cell interaction, immune cell maturation, macrophage activation, T lymphocyte development, differentiation, and cytokine production [67,68,69]. The function of *ly6g6f* in fish is not yet defined, and in this study, the expression of *ly6g6f* was significantly downregulated at 2 wpi in lumpfish larvae (Figure 6 and Appendix A). These results agree with the low level of immune protection.

The *igh* (*immunoglobulin heavy locus*) encodes the IgM heavy chains, and these loci have been characterized in several fish species, including fugu, rainbow trout, zebrafish, and Atlantic salmon [70,71,72]. It has been established that *igh* plays a role during the adaptive immune response by recognizing foreign antigens for phagocytosis, and the complement system [73]. Here, we found that *igha*, *ighd*, and *ighmc* expression were significantly upregulated in orally immunized compared to larvae who had been orally mock immunized at 2 wpi (Figure 5A and Appendix A). These results suggest the oral immunization of larvae triggers some level of an adaptive immune response, but it seems insufficient to trigger memory immune protection. 

*ccl19* is a chemokine known to orchestrate the migration of dendritic cells (DCs) and T cells into lymphoid tissue or vaccination sites and is also involved in immune tolerance and inflammatory responses [74,75]. *ccl20* attracts lymphocytes and DCs towards epithelial cells to mucosal immune sites under inflammatory conditions [76]. In orally immunized compared to larvae who had been orally mock immunized, there was a significant increase in levels of *ccl19* and *ccl20* at 2 wpi, while there was a significant decrease in *ccl20* levels at 4 wpi (Figure 5B and Appendix A). These results aligned with the expression patterns of other transcripts evaluated here, supporting the idea that the lumpfish larvae did respond to the oral immunization.

CD4 (a classical marker of T helper cells) and CD8 (a marker of cytotoxic lymphocytes) are polypeptides that play an important role in signal transduction and the activation of T-helper cells and cytotoxic T cells, respectively [77]. We found that *cd8* was significantly upregulated in orally immunized compared to larvae who had been orally mock immunized at 2 wpi (Figure 5C and Appendix A). However, *cd4* was not significantly dysregulated. These results suggest that CD8 cellular-mediated adaptive immunity, but not the CD4 response, was activated in lumpfish larvae aligning with the lack of immune protection triggered by the oral immunization (Figure 7A). 

CD74 is the MHC class II-associated invariant chain, which plays a role in antigen presentation [78,79]. CD4 and CD74 lost their original functions in anglerfish (*Lophius piscatorius*) and Atlantic cod (*Gadus morhua*) [80,81,82,83]. Here, we found that these transcripts are present in lumpfish, and although *cd74* was upregulated in orally immunized compared to larvae who had been orally mock immunized at 2 wpi, *cd4a* and *cd4b* were not (Figure 5C and Appendix A). These results revealed that oral immunization in lumpfish larvae triggers a partial adaptive immune response. 

The lumpfish larvae were vaccinated after the yolk sac was absorbed, after which the larvae exhibited an active feeding behavior. Although there is no literature about the immunity of lumpfish larvae, it is well known that lumpfish larvae are more mature and active than other marine fish [84]. It has also been shown that the main immune organs of lumpfish develop after hatching [85]. These reports, in addition to our current results, suggest that lumpfish larvae are immune competent, and antigens need to be delivered across the epithelia to trigger full immunity. The transcript expression levels (Figure 4, Figure 5 and Figure 6) also indicated that lumpfish larvae are immune stimulated by oral immunization, but not enough to trigger immune protection. For instance, the expression of *il8b*, *il0, igha*, *ighmc, ighb, cd8,* and *c74* was upregulated in orally immunized larvae (Figure 4, Figure 5 and Figure 6). It seems that oral immunization with *V. anguillarum* bacterin in lumpfish larvae triggered Th1-like immune response and cellular immunity, which is related to *il10* and *cd8* upregulation. This is the first study on lumpfish larvae molecular immunity and provides novel knowledge and a baseline to study the ontogeny of the immune system in lumpfish.

The effectiveness of vaccination in fish depends on the delivery, vaccine design, and the fish species. For instance, mortality in lumpfish bath immunized and i.p. boosted with a commercial polyvalent formalin-inactivated *V. anguillarum* O1 and O2 vaccine was only delayed in an i.p. challenge using *V. anguillarum* [28]. Similar to our current results, a commercial bivalent whole-cell *V. anguillarum* O1 and O2 vaccine delivered by immersion and followed by an i.p. boost immunization in European sea bass (*Dicentrarchus labrax*) conferred approximately 99% survival against a *V. anguillarum* i.p. challenge [86]. In this study, we observed that lumpfish orally immunized as larvae and then orally boosted as juveniles did not survive the *V. anguillarum* i.p. challenge (Figure 7A). Nevertheless, we determined that oral vaccination delayed mortality in lumpfish challenged with *V. anguillarum,* suggesting that the oral vaccination did stimulate fish immunity, but not enough to confer protection. Similar results were found in salmonids orally immunized against *Yersinia* and *V. anguillarum*, where oral immunization conferred no or low immunity to juvenile immunized fish [87,88,89,90]. In contrast, lumpfish orally immunized as larvae and then both orally and i.p. boosted as juveniles showed a significant RPS (76.5%) to the *V. anguillarum* i.p. challenge (Figure 7B). This suggests that the orally administered vaccines were not reaching the deep lymphoid tissues, either in the larvae or juvenile fish, and as such, oral immunization was not effective in contrast to the i.p. delivered vaccine. Therefore, it is suggested that inactivated *V. anguillarum* vaccines for lumpfish should be administered using the i.p. route to confer acceptable levels of immune protection. 

## 5. Conclusions

Oral immunization of lumpfish larvae using bio-encapsulated bacterin demonstrated that it reached the gut and immune, stimulating the fish larvae. However, oral immunization did not trigger an evident adaptive immune response, even after oral boost immunization. *V. anguillarum* bacterin that had been orally administered delayed mortality and did not confer protection against the i.p. *V. anguillarum*. In contrast, i.p. immunization conferred significant immune protection. These results suggest the need for oral vaccines that have the capability of crossing the epithelium and reaching the deep lymphoid tissues to trigger immune protection. 

## Figures and Tables

**Figure 1 vaccines-09-00819-f001:**
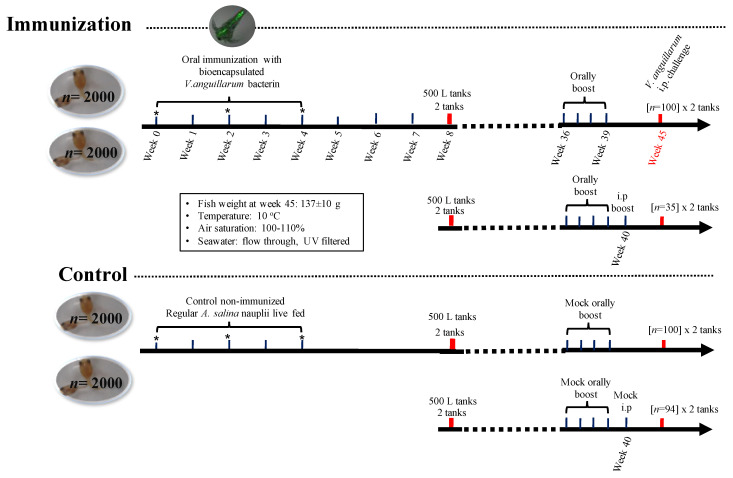
Immunization and challenge experimental design. Lumpfish larvae were immunized at 1 wph. At 8 wph, the fish were distributed in 500 L tanks. Lumpfish were boosted at 36 wph and challenged with *V. anguillarum* at 45 wph. * Larvae samples were collected at weeks 0, 2, 4 post-immunization for RNA extraction.

**Figure 2 vaccines-09-00819-f002:**
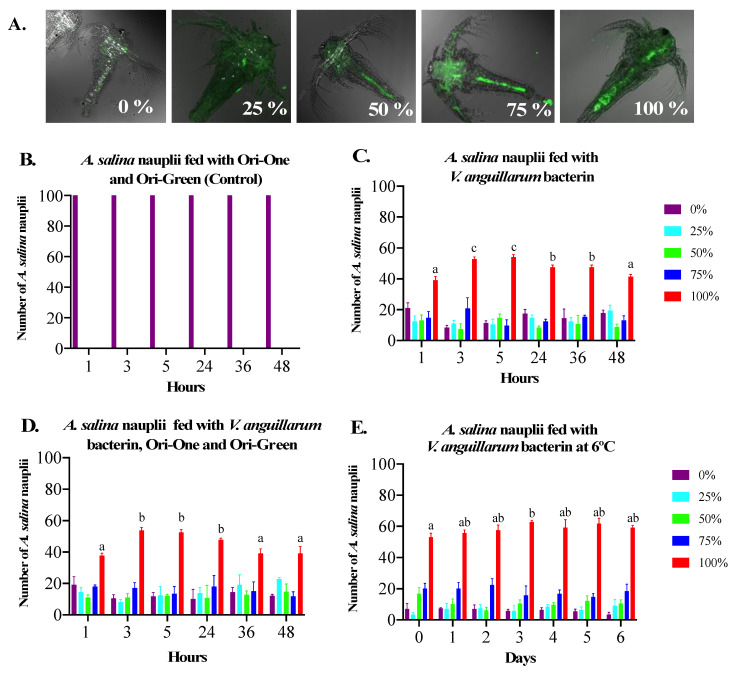
Optimization of *V.*
*anguillarum* bacterin bio-encapsulation in *A. salina* nauplii. (**A**). Relative percentage of DTAF-labeled *V. anguillarum* bacterin in *A. salina* nauplii intestine; (**B**) *A. salina* nauplii suplemeted with dry microalgae (Ori-One and Ori-Green) as autofluorescence control. *A. salina* fed with commercial dry microalgae (Ori-One and Ori-Green) at 20 °C; (**C**) *V. anguillarum* bacterin bio-encapsulation in *A. salina* nauplii at 20 °C; (**D**) *V. anguillarum* bacterin and commercial dry microalgae bio-encapsulation in *A. salina* nauplii at 20 °C; (**E**) bio-encapsulation stability at 6 °C after 3 h post enrichment with *V. anguillarum* bacterin at 20 °C. In all cases, different color bars represent the % bio-encapsulation levels of the *V. anguillarum* bacterin in *A. salina* nauplii. Each value is the mean ± SEM for 3 groups of 100 *A. salina* nauplii per group. Different letters (a, b, c) indicate the differences in the numbers of *A. salina* nauplii enriched with 100% *V. anguillarum* bacterin at different time points. Means with different letters differ significantly (*p* < 0.05). Bars represent mean ± SEM.

**Figure 3 vaccines-09-00819-f003:**
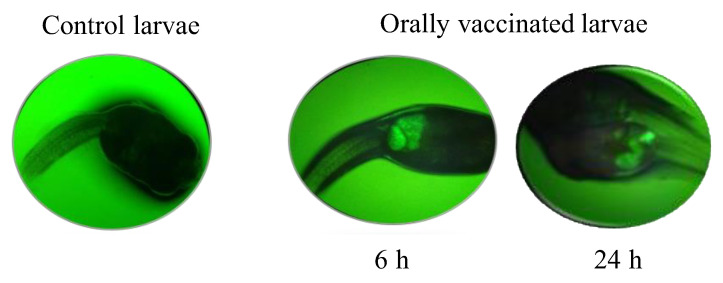
Selected images depicting the DTAF-labeled *V. anguillarum* bacterin bio-encapsulated in *A. salina* nauplii in the lumpfish gut after 6 and 24 h, visualized using epifluorescence microscopy. Its presence is indicated by green fluorescence.

**Figure 4 vaccines-09-00819-f004:**
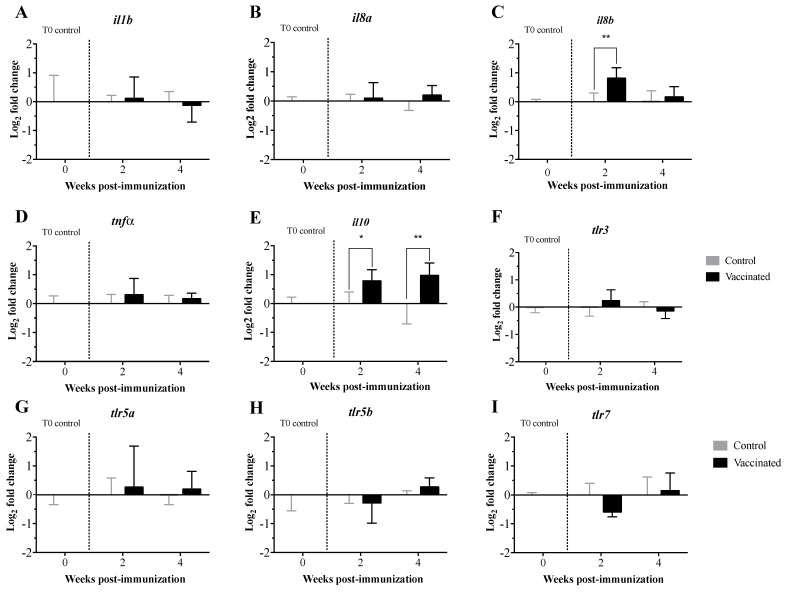
Transcript expression levels of cytokines and toll-like receptors in lumpfish larvae orally immunized with the *V. anguillarum* bacterin bio-encapsulated in *A. salina* nauplii. (**A**–**E**). Cytokines; (**F**–**I**). Toll-like receptors. Transcript expression levels were assessed pre-immunization (T0 control, *n* = 3 individual pools of 10 larvae each), 2 wpi (*n* = 3 pools of 10 larvae each), and 4 wpi (*n* = 3 pools of 5 larvae each). Time point controls post-mock immunization were collected in a similar fashion at 2 and 4 wpi. Relative expression was calculated using the 2^(−∆∆Ct)^ method and normalized using log_2_; *etif3d* and *rpl32* were used as endogenous controls. A two-way ANOVA test, followed by the Sidak multiple comparisons post hoc test was used to assess significant differences between the treatments (control and vaccinated) at each individual time point. Asterisks (*) represent significant differences (* *p* < 0.05, ** *p* < 0.01).

**Figure 5 vaccines-09-00819-f005:**
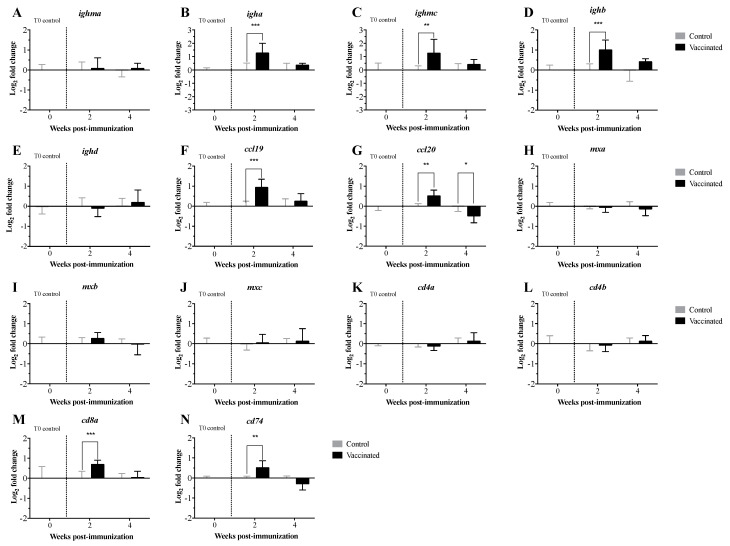
Transcript expression levels of immunoglobulin heavy locus genes, cytokine CC genes and interferon-induced GTP-binding proteins genes, and cluster of differentiation genes in lumpfish larvae orally immunized with the *V. anguillarum* bacterin bio-encapsulated in *A. salina* nauplii (**A**–**E**). Immunoglobulin heavy locus genes (**F**–**J**). Cytokine CC genes and interferon-induced GTP-binding proteins genes (**K**–**N**). Cluster of differentiation genes. Transcript expression levels were assessed pre-immunization (T0 control, *n* = 3 individual pools of 10 larvae each), 2 weeks post-immunization (*n* = 3 pools of 10 larvae each), and 4 weeks post-immunization (*n* = 3 pools of 5 larvae each). Time point controls post-mock immunization were collected in a similar fashion to those collected at weeks 2 and 4. Relative expression was calculated using the 2^(−∆∆Ct)^ method and normalized using log_2_; *etif3d* and *rpl32* were used as endogenous controls. A two-way ANOVA test followed by the Sidak multiple comparisons post hoc test were used to assess significant differences between the treatments (control and vaccinated) at each individual time point. Asterisks (*) represent significant differences (* *p* < 0.05, ** *p* < 0.01, *** *p* < 0.001).

**Figure 6 vaccines-09-00819-f006:**
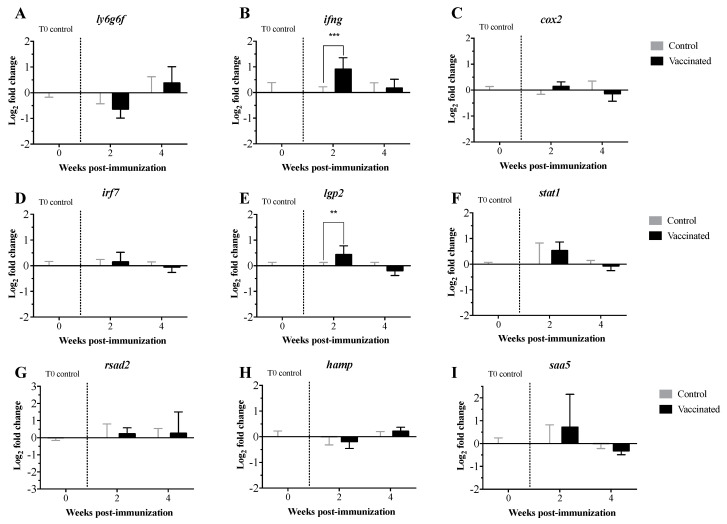
Transcript expression levels of other immune-related genes in lumpfish larvae orally immunized with the *V. anguillarum* bacterin bio-encapsulated in *A. salina* nauplii (**A**–**I**). Transcript expression levels were assessed pre-immunization (T0 control, *n* = 3 individual pools of 10 larvae each), 2 weeks post-immunization (*n* = 3 pools of 10 larvae each), and 4 weeks post-immunization (*n* = 3 pools of 5 larvae each). Time point controls post-mock immunization were collected in a similar fashion to those collected at weeks 2 and 4. Relative expression was calculated using the 2^(−∆∆Ct)^ method and normalized using log_2_; *etif3d* and *rpl32* were used as endogenous controls. A two-way ANOVA test followed by the Sidak multiple comparisons post hoc test were used to assess significant differences between the treatments (control and vaccinated) at each individual time point. Asterisks (*) represent significant differences (** *p*< 0.01, *** *p*< 0.001).

**Figure 7 vaccines-09-00819-f007:**
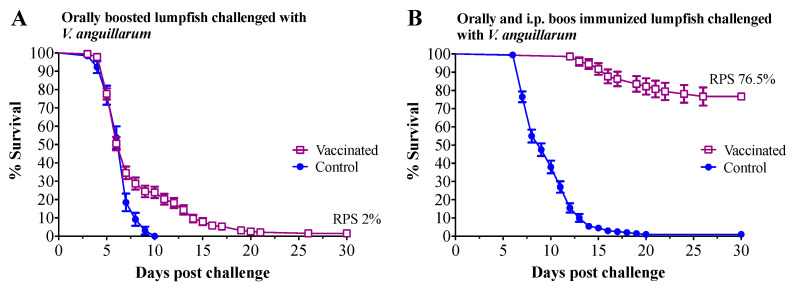
Cumulative survival rate of orally and i.p. immunized lumpfish after i.p. challenge with *V. anguillarum* (7.8 × 10^5^ CFU dose^−1^). (**A**) Survival (%) of orally immunized and orally boosted lumpfish after *V. anguillarum* challenge. Lumpfish were orally immunized as larvae and then orally boosted as juveniles. Control groups were mock-vaccinated using the same inoculation route. After 45 weeks post-initial immunization, the animals were then i.p. challenged; each treatment consisted of two tanks (see Figure 1). (**B**) Survival of orally immunized and i.p. boosted lumpfish. Lumpfish were orally immunized as larvae and then orally boosted as juveniles (see Figure 1). Control groups were mock-vaccinated using the same inoculation route. After 45 weeks post-initial immunization, the animals were then i.p. challenged; Each treatment consisted of two tanks (see Figure 1). Survival was assessed for 30 days. RPS: relative percentage survival; *p* < 0.0001.

## Data Availability

Not applicable.

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
