# Peer review of "Oral Immunization of Larvae and Juvenile of Lumpfish (*Cyclopterus lumpus*) against *Vibrio anguillarum* Does Not Influence Systemic Immunity"

_vaccines, 2021, doi:10.3390/vaccines9080819_

Round 1

Reviewer 1 Report

Review of the manuscript: ‘Larvae and Juvenile Lumpfish (Cyclopterus lumpus) Immunization Against Vibrio anguillarum’.

The first thing to point out is that the title of this article does not attract readers’ interest. I found that the author has studied many indicators in this research, but not all indicators must appear in the title. The author should condense a better title to reflect the highlights of this article. The experimental design is reasonable but lacks innovation, the data is relatively complete, and a lot of modification and editing work are still needed.

The abstract section should briefly introduce the research background and research significance and clarify the research methods, then introduce the main research results, and finally, give the corresponding conclusions.

The material and method section need to be re-integrated and checked to avoid repetition and confuse. All experimental protocols should be better explained. The number of samples used for PCR and biochemical testing is not clear.

There are aspects of the technical presentation that require attention:

Make sure that all works cited in the text are in the reference list, that the presentation is consistent and that correct information is given.

Define and explain all acronyms and abbreviations on first mention in the text.

On first mention of a species in the text give both the common (trivial) and formal name, and make sure that the presentation is correct and consistent.

Make sure that symbols, sub- and super-scripts, upper- and lower-case are presented correctly, and that there is correct and consistent use of italics, brackets and punctuation etc.

There are mistakes in the reference list, including incorrect reporting, inconsistent presentation, spelling mistakes and problems with use of punctuation etc.

Author Response

We appreciate the reviewer's time and suggestion. We feel that the article has improved tremendously. We hope that the current version satisfies the reviewer's requests. 

Reviewer #1

The first thing to point out is that the title of this article does not attract readers’ interest. I found that the author has studied many indicators in this research, but not all indicators must appear in the title. The author should condense a better title to reflect the highlights of this article. The experimental design is reasonable but lacks innovation, the data is relatively complete, and a lot of modification and editing work are still needed.

RE: We modified the title to Oral Immunization of Larvae and Juvenile of Lumpfish (Cyclopterus lumpus) Against Vibrio anguillarum Does Not Influence Systemic Immunity”

The abstract section should briefly introduce the research background and research significance and clarify the research methods, then introduce the main research results, and finally, give the corresponding conclusions.

RE: The abstract was modified to

“Abstract: Vibrio anguillarum, a marine bacterial pathogen that causes vibriosis, is a recurrent pathogen of lumpfish (Cyclopterus lumpus). Lumpfish is utilized as a cleaner fish in the Atlantic salmon (Salmo salar) aquaculture in the North Atlantic region because of its ability to visualize and prey the ectoparasite sea lice (Lepeophtheirus salmonis) on the Atlantic salmon skin and performance in cold environments. Lumpfish immunity is critical for the optimal performance and sea lice removal. Oral vaccine delivery at a young age is the desired method for fish immunization. Oral vaccines are easy to use, reduce fish stress during immunization, and can be applied on a large scale while the fish are at a young age. However, the efficacy of orally delivered inactivated vaccines is controversial. In this study, we evaluated the effectiveness of a V. anguillarum bacterin orally delivered to cultured lumpfish and contrast to an intra peritoneal (i.p.) boost delivery. We bio-encapsulated V. anguillarum bacterin in Artemia salina live-feed and orally immunize lumpfish larvae. Vaccine intake and immune response were evaluated by microscopy and quantitative polymerase chain reaction (qPCR) analysis, respectively. qPCR analyses showed that oral immunization of lumpfish larvae resulted in a subtle stimulation of canonical immune transcripts such as il8b, il10, igha, ighmc, ighb, ccl19, ccl20, cd8a, cd74, infg, and lgp2. Nine months after oral immunization, one group was orally boosted, and a second group was both orally and i.p. boosted. Two months after boost immunization, lumpfish were challenged with V. anguillarum (7.8x105 CFU dose−1). Orally boosted fish showed a relative percentage of survival (RPS) of 2%. In contrast, the oral and i.p. boosted group showed an RPS of 75.5% (p < 0.0001). V. anguillarum bacterin orally delivered was not effective in lumpfish, which in contrast to the i.p. delivered bacterin that protects lumpfish against vibriosis. This suggests that orally administered V. anguillarum bacterin did not reach deep lymphoid tissues, either in the larvae or juvenile fish, therefore oral immunization was not effective. Oral vaccines that are capable to cross the epithelium and reach deep lymphoid tissues are required to confer effective protection to lumpfish against V. anguillarum.”

The material and method section need to be re-integrated and checked to avoid repetition and confuse. All experimental protocols should be better explained. The number of samples used for PCR and biochemical testing is not clear.

RE: All the material and method section was revised and modified. The number of samples for PCR also was clearly stated.

Total RNA extraction. RNA was extracted form larvae pools pre-immunized (n= 3 individual pools of 10 larvae each), 2 wpi (n= 3 pools of 10 larvae each), and 4 wpi (n= 3 pools of 5 larvae each).”

There are aspects of the technical presentation that require attention:

Make sure that all works cited in the text are in the reference list, that the presentation is consistent and that correct information is given.

RE: All the references were revised and modified for constancy.

Define and explain all acronyms and abbreviations on first mention in the text.

RE: All acronyms and abbreviations were defined in the first mention.

On first mention of a species in the text give both the common (trivial) and formal name, and make sure that the presentation is correct and consistent.

RE: All the species and common names were mentioned as requested.

Make sure that symbols, sub- and super-scripts, upper- and lower-case are presented correctly, and that there is correct and consistent use of italics, brackets and punctuation etc.

RE: All the aspects were fixed and checked.

There are mistakes in the reference list, including incorrect reporting, inconsistent presentation, spelling mistakes and problems with use of punctuation etc.

RE: All the reference formats and lists were updated.

Reviewer 2 Report

The topic of this article is very interesting since, in most of the cases, infectious diseases are only studied to be remediated in fish species destined to human consumption whilst not in that which acts as vectors. The evaluation of vaccination methods against V. anguillarum in lumpfish, that is used as a cleaner fish to biocontrol ectoparasitic infestations in Atlantic salmon would avoid later infections. It is a high-quality manuscript, very complete with a robust analysis. However, I have a major concerned and insurmountable obstacle based on the experimental design and discussion of the data obtained.

In this article, lumpfish larvae have been vaccinated with complete antigen, V. anguillarum bacterin. However, larvae first vaccination is performed after only 1 week post-hatching. The authors did not include bibliography regarding ontogenetic immune development of lumpfish and, to the best of my knowledge, it is still unknown when the immune system of lumpfish is fully functional to not generate tolerance to antigens (as vaccines). In teleost fish, ontogeny of the immune system is age- and species-dependent but fish usually rise mature levels of antibodies and other adaptive immunity after several weeks after hatching (very often after at least 50-60 days post-hatching or larger periods) depending of the fish species. At this regard, vaccinating 1 week larvae would almost certainly generate tolerance to the vaccine. In fact, authors claim at several times during the discussion that their data demonstrate the immaturity of the immune system of the larvae used (e.g. lines 483-485; 507-508). As matter of fact, a vaccine applied prior to reach complete immunocompentence  could also lead into a lesser immune response in later stages of development.

Misleading vaccine tolerance triggered, the larvae could improve survival upon challenge, as seen in this manuscript results but probably due to the enhancement of innate responses mechanisms. However, it been ignored the tolerance mechanisms generation.

This is a major problem in the experimental design of vaccines development so I recommend to reject the manuscript.

Regarding formal aspects of the manuscript:

Who owns filiation number 3?

Line 48: “green alternative” is an ambiguous and subjective term, authors should avoid using it.

Line 85: Lumpfish groups were orally boosted and then orally and ip boosted? Or were they orally or orally and ip boosted (as two different groups)?

In supplementary figure S3 there are two green micrographies but into them, it is not clarified what we have to observe (although it is kind of intuitive it should be clarified).

To coat the commercial dry feed with dry V. anguillarum bacterin authors used Ficoll solution. How they think it would affect the bacteria? Have them studied possible toxicity events? In some concentrations depending on the cell type Ficoll could be toxic.

The code of the Ethical Committee approval must be included, or are they the protocol codes # that have been included?

English should be revised (minor mistakes review).

The point 2.7. should be titled only Fish culture conditions since immunization is described in later points.

Point 2.8. Lumpfish immunization assays. Regarding control group, I understand that all mock-boosters were performed with Ficoll as it is the bacteria vehicle. However, did the authors performed any assay to compare what happened with a mock-vaccinated group without the vehicle? Using PBS or culture medium e.g. It has been described that many vaccines vehicles can trigger considerably immune responses in fish so it is very important to include a strict control group in vaccination assays. Are there any information available regarding immune responses elicited by Ficoll?

Figure 2:

  • Which is Fig. 2B? it is not described in results (point 3.1 at least). Is it a graphic of controls with or without A. salina? Or if its only with the Ori-one and Ori-Green additive why there is a group with 100% of A. salina? It is not clear the legend or the concept of this graphic. However, since only A. salina a were detected for the 100%, for the rest of the groups should record ND (non-detected) or similar to clarify that all groups have been measured. Since the test was run in triplicates and there is no error bar, it is assuming that all replicates showed a minimal or non error?
  • Is the legend of Fig. 2B the same for 2C-E? it should be clarified.

For transcriptional levels analysis (Fig. 4-6) figures should show the relative expression instead the fold change since most of the bars are not shown in the current format, and also at time 0 there is only one group analysed so the fold change does not exist and comparison has no sense.

Figure 7: Although it could be a bit intuitive, I don’t really understand the description of the figure and so the results showed since there are two different descriptions for Fig. 7A and two for Fig. 7B. Description for each figure should be concreted in only one each.

Author Response

Reviewer #2

The topic of this article is very interesting since, in most of the cases, infectious diseases are only studied to be remediated in fish species destined to human consumption whilst not in that which acts as vectors. The evaluation of vaccination methods against V. anguillarum in lumpfish, that is used as a cleaner fish to biocontrol ectoparasitic infestations in Atlantic salmon would avoid later infections. It is a high-quality manuscript, very complete with a robust analysis. However, I have a major concerned and insurmountable obstacle based on the experimental design and discussion of the data obtained.

In this article, lumpfish larvae have been vaccinated with complete antigen, V. anguillarum bacterin. However, larvae first vaccination is performed after only 1 week post-hatching. The authors did not include bibliography regarding ontogenetic immune development of lumpfish and, to the best of my knowledge, it is still unknown when the immune system of lumpfish is fully functional to not generate tolerance to antigens (as vaccines). In teleost fish, ontogeny of the immune system is age- and species-dependent but fish usually rise mature levels of antibodies and other adaptive immunity after several weeks after hatching (very often after at least 50-60 days post-hatching or larger periods) depending of the fish species. At this regard, vaccinating 1 week larvae would almost certainly generate tolerance to the vaccine. In fact, authors claim at several times during the discussion that their data demonstrate the immaturity of the immune system of the larvae used (e.g. lines 483-485; 507-508). As matter of fact, a vaccine applied prior to reach complete immunocompentence  could also lead into a lesser immune response in later stages of development.

Misleading vaccine tolerance triggered, the larvae could improve survival upon challenge, as seen in this manuscript results but probably due to the enhancement of innate responses mechanisms. However, it been ignored the tolerance mechanisms generation.

This is a major problem in the experimental design of vaccines development so I recommend to reject the manuscript.

RE:

We appreciate the reviewer's candour and valuable feedback for improvement. We did in fact vaccinate the lumpfish larvae after the yolk sacs were adsorbed, and after the larvae exhibited positive feeding behavior. Although there is currently no literature available that speaks specifically to immunity of lumpfish larvae, it is well known that lumpfish larvae are more mature and active compared to other marine fish. Also, it has been shown that the main immune organs in lumpfish have developed post-hatch. We have added additional in-text citations/references to address this aspect of the reviewer's comments. Also, as that this is the first study on lumpfish larvae immunization, the current study provides novel knowledge and a baseline to study the ontogeny of the immune system in lumpfish. 

            We did not discuss the aspect of immune tolerance because is not the scope of the current article, in addition, we provide with a high dose of antigen, not a low dose that could trigger oral tolerance. Also, the concept of “in ovo” vaccination is applied to fish and birds. Commercial vaccines against viruses are applied directly to the chicken egg successfully in the poultry industry and they have been experimentally described in fish. The concept of oral tolerance is not clear in lumpfish. In addition, the i.p. boosted group had an RPS of 76% indicating that the oral immunization did not cause vaccine tolerance.  The reviewer made very important comments and we realized that the immune markers utilized indicated that lumpfish larvae are immune-competent. However, oral immunization with inactivated antigens is not triggering protection.

The manuscript was modified as:

Line 476-505:

“The lumpfish larvae were vaccinated after the yolk sac was absorbed, and after the larvae exhibited an active feeding behaviour. Although there is no literature about the immunity of lumpfish larvae, it is well known that lumpfish larvae are more mature and active compared to other marine fish [84]. Also, it has been shown that the main immune organs of lumpfish have been developed after hatch [85]. These reports, in addition to our current results, suggest that lumpfish larvae are immune competent, and antigens need to be delivered across epithelia to trigger a full immunity.

Oral administration of low levels of antigens could trigger vaccine tolerance in fish and can increase susceptibility to the target pathogen [86]. Here, we orally administrate high concentration of antigens to prevent possible oral tolerance. Also, the transcript expression levels (Figs. 4-6) indicated that lumpfish larvae are immune stimulated by the oral immunization, but not enough for triggering immune protection. In addition, i.p. boosted lumpfish show a high immunity against V. anguillarum challenge and discarding induction of vaccine tolerance. Also, the transcription expression analysis did not suggest an oral tolerance profile. For instance expression of il8b, il0, igha, ighmc, ighb, cd8, and c74 was upregulated in orally immunized larvae (Figs. 4-6), in contrast to oral tolerance expression profile where il10 is down-regulated [86]. It seems that oral immunization with V. anguillarum bacterin in lumpfish larvae triggered Th1-like immune response and cellular immunity, related to il10 and cd8 up-regulation, respectively. This is the first study on lumpfish larvae molecular immunity and provides novel knowledge and a baseline to study the ontogeny of the immune system in lumpfish.

The effectiveness of vaccination in fish depends on the delivery, vaccine design, and fish species. For instance, mortality in lumpfish bath immunized and i.p. boosted with a commercial polyvalent formalin‐inactivated V. anguillarum O1 and O2 vaccine only was delayed in an i.p. challenge with V. anguillarum [28]. Similar to our results current results, a commercial bivalent whole-cell V. anguillarum O1 and O2 vaccine delivered by immersion and followed by an i.p. boost immunization in European sea bass (Dicentrarchus labrax) conferred approximately 99% survival against V. anguillarum i.p. challenge [87]. In this study, we observed that lumpfish orally immunized as larvae and then orally boosted as juveniles, did not survive the V. anguillarum i.p. challenge (Fig. 7A). Nevertheless, we determined that oral vaccination delayed mortality in lumpfish challenged with V. anguillarum, suggesting that the oral vaccination stimulated fish immunity but not enough to confer protection. Similar results were found in salmonids orally immunized against Yersinia and V. anguillarum, where oral immunization conferred no or low immunity to juvenile immunized fish [88-91]. In contrast, lumpfish orally immunized as larvae, and then both orally and i.p. boosted as juveniles showed a significant RPS (76.5%) to the V. anguillarum i.p. challenge (Fig. 7B). This suggests that orally administered vaccines were not reaching deep lymphoid tissues, either in the larvae or juvenile fish, and as such, oral immunization was not effective, in contrast to the i.p. delivered vaccine.”

Regarding formal aspects of the manuscript:

Who owns filiation number 3?

RE: The text was modified to :

My Dang1, Trung Cao1, Ignacio Vasquez1, Ahmed Hossain1, Hajarooba Gnanagobal1, Surendra Kumar2, Jennifer R Hall3, Jennifer Monk4, Danny Boyce4, Jillian Westcott5, Javier Santander1*

1     Marine Microbial Pathogenesis and Vaccinology Laboratory, Department of Ocean Sciences, Memorial University of Newfoundland, St. John's, NL, Canada; tmtdang@mun.ca (M.D.); (I.V) ivasquezsoli@mun.cattcao@mun.ca (T.C.); ahossain@mun.ca (A.H.); hgnanagobal@mun.ca (H.G.)

2    Ocean Frontier Institute, Department of Ocean Sciences, Memorial University of Newfoundland, St. John's, NL, Canada; surendrak@mun.ca (S.K.)

3     Aquatic Research Cluster, CREAIT Network, Department of Ocean Sciences, Memorial University of Newfoundland, St. John's, NL, Canada jrhall@mun.va (J.H.)

4   Dr. Joe Brown Aquatic Research Building (JBARB), Memorial University of Newfoundland, St. John's, NL, Canada; jmonk@mun.ca (J.M.); dboyce@mun.ca (D.B.)

5   Fisheries and Marine Institute of Memorial University of Newfoundland, St. John's, NL, Canada; jillian.westcott@mi.mun.ca (J.W.)

Line 48: “green alternative” is an ambiguous and subjective term, authors should avoid using it.

RE: The sentence was modified to “The use of cleaner fish is considered an eco-friendly alternative as it can reduce the use of chemotherapeutants and it is less stressful to farmed fish

Line 85: Lumpfish groups were orally boosted and then orally and ip boosted? Or were they orally or orally and ip boosted (as two different groups)?

RE: The sentence was modified to “Nine months later, a lumpfish group was orally boosted, and an independent second group was orally and i.p boosted.”

In supplementary figure S3 there are two green micrographies but into them, it is not clarified what we have to observe (although it is kind of intuitive it should be clarified).

RE: The figure and legend were modified to:

Figure S3. Oral immunization of lumpfish larvae with the DTAF-labeled V. anguillarum bacterin bio-encapsulated in Artemia nauplii. Lumpfish larvae (7 dph) were either fed Artemia nauplii with the bio-encapsulated DTAF-labeled V. anguillarum bacterin or control Artemia nauplii that had been inoculated with seawater, and maintained at 6 °C for 12 h. The presence of V. anguillarum bacterin in the gut of lumpfish larvae compared to non-orally immunized fish was then assessed at 0, 0.5, 1, 2, 4, and 6 h post-oral immunization by fluorescence microscopy. The arrow indicates the presence of green-fluorescent V. anguillarum bacterin.

To coat the commercial dry feed with dry V. anguillarum bacterin authors used Ficoll solution. How they think it would affect the bacteria? Have them studied possible toxicity events? In some concentrations depending on the cell type Ficoll could be toxic.

RE: Ficoll is a standard non-toxic polymer utilized for many applications, bacterial lyophilization, bacterin preparations, and immunological studies. Although we did not used alive bacteria with Ficoll, there are several studies for bacterial preservation using Ficoll formulations. Also, Ficoll has been used in trout without side effects. The specific effect of ficoll in lumpfish have not been studied and this is the first time that is used in this fish species. However, this is not the focus of this particular article, and the effect of Ficoll on protective immunity can be rule out by the strict control treatment (Ficoll coated pellets).

For clarification we added references and modified the text to: Lines 155-171 “Aquafeed coating with V. anguillarum bacterin. Commercial dry feed was coated with dry V. anguillarum bacterin to orally boost immunized fish. Ficoll, a non-toxic polymer, was utilized as a cryoprotectant for bacterial lyophilization [41]. Also, Ficoll serves as antigen and adjuvant carrier [41-46]. To freeze-dry the bacterin, a formalin-killed V. anguillarum (2x109 CFU mL−1) suspension was mixed with Ficoll solution (20% Ficoll400 (GE Healthcare, Sweden), 300 mM NaCl) at a 1:1 ratio to prevent cell lysis during lyophilization. The cells were lyophilized (Edwards super module E2-M5, Edwards, UK) for 3 days. The bacterin powder was mixed with 3-4 mm commercial dry pellet (Skretting-Europa 15: crude protein (55%), crude fat (15%), crude fiber (1.5%), calcium (3%), phosphorus (2%), sodium (1%), vitamin A (5,000 IU/kg), vitamin D (3,000 IU/kg) and vitamin E (200 IU/kg)) at the ratio of 0.9 g bacterin per 100 g aquafeed. After mixing the feed with the bacterin powder, a layer of cod liver oil was added (3 mL 100 g feed-1) and the feed was then dried at room temperature to complete the coating process. The coated feed was stored at 4°C until utilization.

The code of the Ethical Committee approval must be included, or are they the protocol codes # that have been included?

RE: These are in original manuscript: See lines 161-163: “All animal protocols required for this research were approved by the Institutional Animal Care Committee and the Biosafety Committee at Memorial University of Newfoundland (MUN). Experiments were conducted under protocols #18-01-JS, #18-03-JS, and biohazard license L-01” and lines 570-572: “Institutional Review Board Statement: Animal protocols #18-01-JS, #18-03-JS (21 Jan 2018), and biohazard licence L-01, were reviewed and approved by the Institutional Animal Care Committee (https://www.mun.ca/research/about/acs/acc/) following the Canadian Council of Animal Care guidelines (https://www.ccac.ca/).

English should be revised (minor mistakes review).

The point 2.7. should be titled only Fish culture conditions since immunization is described in later points.

RE: The text was modified to “Fish culture conditions”

Point 2.8. Lumpfish immunization assays. Regarding control group, I understand that all mock-boosters were performed with Ficoll as it is the bacteria vehicle. However, did the authors performed any assay to compare what happened with a mock-vaccinated group without the vehicle? Using PBS or culture medium e.g. It has been described that many vaccines vehicles can trigger considerably immune responses in fish so it is very important to include a strict control group in vaccination assays. Are there any information available regarding immune responses elicited by Ficoll?

RE: We concur that it is essential to include strict control in vaccination assays. Also, we agree with the reviewer's suggestion to use the bacterin suspension media as a control. However, it was exactly what we did here. The bacterin was lyophilized in a Ficoll suspension (see section 2.6). This powder was mixed with cod oil to coat the aquafeed pellets. Thus, the strict control for the current assays is aquafeed pellets coated with the same suspension media in the absence of the bacterin. In this study, we did not use culture media for fish treatment or vaccine preparation. Therefore, this is not a control since we cannot compare it with any of the vaccine treatments. In addition, it has been described for many fish species, including lumpfish (Chakraborty et al., 2019; Vasquez et al., 2020), that PBS does not trigger immune protection against V. anguillarum. We found that Ficoll orally administered to the lumpfish did not trigger protection against V. anguillarum (see Figure 7). This result rules out the possible effect of Ficoll on adaptive immunity. 

Figure 2:

  • Which is Fig. 2B? it is not described in results (point 3.1 at least). Is it a graphic of controls with or without A. salina? Or if its only with the Ori-one and Ori-Green additive why there is a group with 100% of A. salina? It is not clear the legend or the concept of this graphic. However, since only A. salina a were detected for the 100%, for the rest of the groups should record ND (non-detected) or similar to clarify that all groups have been measured. Since the test was run in triplicates and there is no error bar, it is assuming that all replicates showed a minimal or non error?

RE: The figure 2B is the autofluorescence control, and should indicate 0%. We fixed this error. The figure 2 and the legend were also modified. Yes, there is no error.

  • Is the legend of Fig. 2B the same for 2C-E? it should be clarified.

RE: No, the text was modified to:

Figure 2. Optimization of V. anguillarum bacterin bio-encapsulation in A. salina nauplii. A. Percentage of DTAF-labeled V. anguillarum bacterin in A. salina nauplii intestine; B. Autofluorescence control. A. salina fed with commercial dry microalgae (Ori–One and Ori–Green) at 20°C; C. V. anguillarum bacterin bio-encapsulation in A. salina nauplii at 20°C; D. V. anguillarum bacterin and commercial dry microalgae bio-encapsulation in A. salina nauplii at 20°C; E. Bio-encapsulation stability at 6°C. In all cases, different color bars represent the % bio-encapsulation levels of the V. anguillarum bacterin in A. salina nauplii. Each value is the mean ± SEM for 3 groups of 100 A. salina nauplii per group. Different letters (a, b, c) indicated the differences in the numbers of A. salina nauplii enriched with 100% V. anguillarum bacterin at different time points. Means with different letters differ significantly (p<0.05). Bars represent mean +/- SEM.

For transcriptional levels analysis (Fig. 4-6) figures should show the relative expression instead the fold change since most of the bars are not shown in the current format, and also at time 0 there is only one group analysed so the fold change does not exist and comparison has no sense.

RE: The RQ is values are shown in supplementary figures S5-S7. No significant differences were noticed in these graphs in comparison with the current figures. The Ct values and the relative expression also is shown in supplementary tables S2 and S3.  We did no compare fold change at T0 as there is no treatment. We have this time point as a reference only. We compare the fold changes only at 2 and 4 wpi between specific time points. To clarify these points we modify the figures and the legends to:

Figure 4. Transcript expression levels of cytokines and toll-like receptors in lumpfish larvae orally immunized with the V. anguillarum bacterin bio-encapsulated in A. salina nauplii. A-C. Cytokines; E-H. Toll-like receptors. Transcript expression levels were assessed pre-immunization (T0 control, n= 3 individual pools of 10 larvae each), 2 wpi (n= 3 pools of 10 larvae each), and 4 wpi (n= 3 pools of 5 larvae each). Time point controls post-mock immunization were collected in a similar fashion 2 and 4 wpi. Relative expression was calculated using the 2(−∆∆Ct) method and normalized using log2; etif3d and rpl32 were used as endogenous controls. A two-way ANOVA test, followed by Sidak multiple comparisons post hoc test was used to assess significant differences between the treatments (control and vaccinated) at each individual time point. Asterisks (*) represent significant differences (*p< 0.05, **p< 0.01).

Figure 5. Transcript expression levels of immunoglobulin heavy locus genes, cytokine CC genes and interferon-induced GTP-binding proteins genes, and cluster of differentiation genes in lumpfish larvae orally immunized with the V. anguillarum bacterin bio-encapsulated in A. salina nauplii. A-E. Immunoglobulin heavy locus genes; F-J. Cytokine CC genes and interferon-induced GTP-binding proteins genes; K-N. Cluster of differentiation genes. Transcript expression levels were assessed pre-immunization (T0 control, n= 3 individual pools of 10 larvae each), 2 weeks post-immunization (n= 3 pools of 10 larvae each), and 4 weeks post-immunization (n= 3 pools of 5 larvae each). Time point controls post-mock immunization were collected in a similar fashion for weeks 2 and 4. Relative expression was calculated using the 2(−∆∆Ct) method and normalized using log2; etif3d and rpl32 were used as endogenous controls. A two-way ANOVA test, followed by Sidak multiple comparisons post hoc test was used to assess significant differences between the treatments (control and vaccinated) at each individual time point. Asterisks (*) represent significant differences (**p< 0.01, ***p< 0.001).

Figure 6. Transcript expression levels of other immune-related genes in lumpfish larvae orally immunized with the V. anguillarum bacterin bio-encapsulated in A. salina nauplii (A-J). Transcript expression levels were assessed pre-immunization (T0 control, n= 3 individual pools of 10 larvae each), 2 weeks post-immunization (n= 3 pools of 10 larvae each), and 4 weeks post-immunization (n= 3 pools of 5 larvae each). Time point controls post-mock immunization were collected in a similar fashion for weeks 2 and 4. Relative expression was calculated using the 2(−∆∆Ct) method and normalized using log2; etif3d and rpl32 were used as endogenous controls. A two-way ANOVA test, followed by Sidak multiple comparisons post hoc test was used to assess significant differences between the treatments (control and vaccinated) at each individual time point. Asterisks (*) represent significant differences (**p< 0.01, ***p< 0.001).

Figure 7: Although it could be a bit intuitive, I don’t really understand the description of the figure and so the results showed since there are two different descriptions for Fig. 7A and two for Fig. 7B. Description for each figure should be concreted in only one each.

RE: Figure 7 was modified to

Figure 7. Cumulative survival rate of orally and i.p. immunized lumpfish after i.p. challenge with V. anguillarum (7.8x105 CFU dose-1). A. Survival (%) of orally immunized and orally boosted lumpfish after V. anguillarum challenge. Lumpfish were orally immunized as larvae and then orally boosted as juveniles. Control groups were mock-vaccinated using the same inoculation route. Then, after 45 weeks post-initial immunization the animals were i.p. challenged; Each treatment consisted of two tanks (see Fig. 1). B. Survival of orally immunized and i.p. boosted lumpfish. Lumpfish was orally immunized as larvae and then, orally and i.p. boosted as juvenile (see Fig. 1). Control groups were mock-vaccinated using the same inoculation route. Then, after 45 weeks post-initial immunization the animals were i.p. challenged; Each treatment consisted of two tanks (see Fig. 1). Survival assessed for 30 days. RPS: relative percentage survival; p < 0.0001.

Round 2

Reviewer 2 Report

I have read the manuscript and also the authors review. Although the authors claim that the development of tolerance to the vaccine is not the scope of the manuscript (and it is true that it is not), when applying a vaccine, most likely whole-pathogen based on, implies undeniably the administration to immunocompetent individuals as a first rule. On this basis, you can administer immunomodulators that do not generate immunotolerance, but not a whole-pathogen vaccine without corroborate the concrete age at which specimens are fully immunocompetence. In fact, authors ascertain that “there is no literature about the immunity of lumpfish larvae”, the fact that the immune organs are developed does not mean that they are fully competent and they could rise a fully and mature adaptive response, so the possibility of tolerance development is still a fact.

Another contradicting conclusion given by the authors is that in the first version they claimed that the fish used were not immunocompetent according to their results. In this new version they claimed that larvae are immunocompetent. However, they did not study at any stage of their study the ontogeny of the lumpfish immune system. In this sense, the conclusion is not based on their study.

Based on those two important axis for vaccine development, which are not accomplished with this experimental design, and despite that in my first review I included some recommendations for the formal aspects of the manuscript and authors followed, I cannot recommend this manuscript to be accepted and should be rejected.

Author Response

Rebuttal: Larvae and Juvenile Lumpfish (Cyclopterus lumpus) Immunization Against Vibrio anguillarum

Reviewer #2

I have read the manuscript and also the authors review. Although the authors claim that the development of tolerance to the vaccine is not the scope of the manuscript (and it is true that it is not), when applying a vaccine, most likely whole-pathogen based on, implies undeniably the administration to immunocompetent individuals as a first rule. On this basis, you can administer immunomodulators that do not generate immunotolerance, but not a whole-pathogen vaccine without corroborate the concrete age at which specimens are fully immunocompetence. In fact, authors ascertain that “there is no literature about the immunity of lumpfish larvae”, the fact that the immune organs are developed does not mean that they are fully competent and they could rise a fully and mature adaptive response, so the possibility of tolerance development is still a fact.

Another contradicting conclusion given by the authors is that in the first version they claimed that the fish used were not immunocompetent according to their results. In this new version they claimed that larvae are immunocompetent. However, they did not study at any stage of their study the ontogeny of the lumpfish immune system. In this sense, the conclusion is not based on their study.

Based on those two important axis for vaccine development, which are not accomplished with this experimental design, and despite that in my first review I included some recommendations for the formal aspects of the manuscript and authors followed, I cannot recommend this manuscript to be accepted and should be rejected.

RE:

We appreciate the reviewer's candour. Again, we did in fact vaccinate the lumpfish larvae after the yolk sacs were adsorbed, and after the larvae exhibited positive feeding behaviour. Although there is currently no literature available that speaks specifically to immunity of lumpfish larvae, it is well known that lumpfish larvae are more mature and active compared to other marine fish. Also, it has been shown that the main immune organs in lumpfish have developed post-hatch. Also, as that this is the first study on lumpfish larvae immunization, the current study provides novel knowledge and a baseline to study the ontogeny of the immune system in lumpfish. Here, we did not study the ontogeny of the immune system of lumpfish.  

Although the reviewer claimed “the fact that the immune organs are developed does not mean that they are fully competent and they could rise a fully and mature adaptive response, so the possibility of tolerance development is still a fact”. First, again, we did not develop an experimental design to study immune tolerance. According to the literature, tolerance could be induced by providing low levels of antigens for a prolonged period of time. Actually, we provided a high concentration of antigen for a short period of time. Second, we did not intend to study the ontogeny of the lumpfish larvae, but this is the first study on lumpfish larvae immune stimulation. Finally, the i.p. boosted group had an RPS of 76% indicating that the oral immunization did not cause vaccine tolerance.  The reviewer made very important comments and we realized that the immune markers utilized indicated that lumpfish larvae are immune-competent. Form the revised article indicated: Lines 474-476: “For instance, expression of il8b, il0, igha, ighmc, ighb, cd8, and c74 was upregulated in orally immunized larvae (Figs. 4-6), in contrast to oral tolerance expression profile where il10 is down-regulated [86]. It seems that oral immunization with V. anguillarum bacterin in lumpfish larvae triggered Th1-like immune response and cellular immunity, related to il10 and cd8 up-regulation, respectively. This is the first study on lumpfish larvae molecular immunity and provides novel knowledge and a baseline to study the ontogeny of the immune system in lumpfish”. There is no contradiction and we appreciated the reviewers’ comments.

Also, as we mentioned previously, the concept of “in ovo” vaccination is applied to fish and birds. Commercial vaccines against viruses are applied directly to the chicken egg successfully in the poultry industry and they have been experimentally described in fish. At this time, the concept of oral tolerance is not clear in lumpfish, but we did not observe tolerance in this study, since the i.p. boosted group had an RPS of 76% to the i.p. V. anguillarum challenge, indicating that the oral immunization did not cause vaccine tolerance.  
